# New Approach Methods for Hazard Identification: A Case Study with Azole Fungicides Affecting Molecular Targets Associated with the Adverse Outcome Pathway for Cholestasis

**DOI:** 10.3390/cells11203293

**Published:** 2022-10-19

**Authors:** Constanze Knebel, Roderich D. Süssmuth, Helen S. Hammer, Albert Braeuning, Philip Marx-Stoelting

**Affiliations:** 1Department Food Safety, German Federal Institute for Risk Assessment, Max-Dohrn-Street 8-10, 10589 Berlin, Germany; 2Institute of Chemistry, Technical University Berlin, Straße des 17. Juni 124, 10623 Berlin, Germany; 3Signatope GmbH, Markwiesenstrasse 55, 72770 Reutlingen, Germany; 4Department Pesticides Safety, German Federal Institute for Risk Assessment, Max-Dohrn-Street 8-10, 10589 Berlin, Germany

**Keywords:** hepatotoxicity, azole fungicides, molecular targets, adverse outcome pathway, liver cholestasis

## Abstract

Triazole fungicides such as propiconazole (Pi) or tebuconazole (Te) show hepatotoxicity in vivo, e.g., hypertrophy and vacuolization of liver cells following interaction with nuclear receptors such as PXR (pregnane-X-receptor) and CAR (constitutive androstane receptor). Accordingly, azoles affect gene expression associated with these adverse outcomes in vivo but also in human liver cells in vitro. Additionally, genes indicative of liver cholestasis are affected in vivo and in vitro. We therefore analyzed the capability of Pi and Te to cause cholestasis in an adverse outcome pathway (AOP)-driven approach in hepatic cells of human origin in vitro, considering also previous in vivo studies. Bile salt export pump (BSEP) activity assays confirmed that both azoles are weak inhibitors of BSEP. They alternate the expression of various cholestasis-associated target genes and proteins as well as the mitochondrial membrane function. Published in vivo data, however, demonstrate that neither Pi nor Te cause cholestasis in rodent bioassays. This discrepancy can be explained by the in vivo concentrations of both azoles being well below their EC50 for BSEP inhibition. From a regulatory perspective, this illustrates that toxicogenomics and human in vitro models are valuable tools to detect the potential of a substance to cause a specific type of toxicity. To come to a sound regulatory conclusion on the in vivo relevance of such a finding, results will have to be considered in a broader context also including toxicokinetics in a weight-of-evidence approach.

## 1. Introduction

Cholestasis is a form of substance-induced liver injury that results from an impairment of bile acid excretion causing accumulation of bile acids in the liver and/or the systemic circulation [1]. There are several potential causes of cholestasis, like obstruction of the bile duct, hepatic inflammation or drug–drug interactions [1]. At the molecular level, inhibition of the bile salt export pump (BSEP) is frequently considered as the molecular initiating event (MIE) of the AOP for liver cholestasis, leading to an increase in cellular bile acids and subsequent toxicity [2,3]. Inhibition of BSEP is a frequent cause of substance-induced cholestasis [4]. Several nuclear receptors are involved in bile acid-dependent signaling, especially the constitutive androstane receptor (CAR), the pregnane-X-receptor (PXR), and the farnesoid-X-receptor (FXR) that is activated upon accumulation of bile acids and regulates a number of genes important for bile acid detoxification [5]. The induction of detoxification can be described as an adaptive response, while other events including the induction of mitochondrial dysfunction and oxidative stress can be described as deteriorative responses [6]. A schematic overview is given in Figure 1a that proposes a modified AOP for substance-induced cholestasis based on Vinken et al., 2015 [2].

Triazoles are frequently used fungicides, which display hepatotoxicity in animal studies in rats or mice [7,8]. Hepatocellular hypertrophy is among the most prominent histopathological findings after repeated-dose administration of triazole fungicides, and probably related to activation of a set of nuclear receptors including PXR and CAR, as recently reviewed by Marx-Stoelting et al., 2020 [9]. Some fungicides of this group have also been shown to cause hepatocellular cholestasis: prothioconazole, for example, was reported to be cholestatic in respective in vivo studies conducted for its approval as an active substance for use in pesticides [10]. Moreover, cholestasis is a frequently described finding in drug-induced liver injury caused by pharmacologically used azoles [11]. The ability of azole fungicides to interact with BSEP, the MIE of cholestasis (Figure 1), has not been investigated so far for a number of triazoles, including tebuconazole (Te) and propiconazole (Pi). Since hepatic gene expression potentially related to cholestasis was altered by azoles such as Pi, Te or cyproconazole in previous studies involving transcriptomics analysis in vivo or in vitro [12,13], we decided to further elucidate the potential of Pi and Te to cause cholestasis by application of new approach methods (NAMs). NAMs are under development in several projects for the identification of different forms of hepatotoxicity [12,13,14]. Such methods are considered to help in identifying human-relevant liver effects while at the same time avoiding animal testing. Central question are how well AOP-based batteries of NAMs are capable of predicting adverse effects in vivo, and how well findings are correlated with human disease. For the endpoint liver steatosis, we were recently able to show that this correlation is quite pronounced [15]. For the endpoint cholestasis, the work of others points in the same direction [3].

The aim of our study was to investigate mechanisms of triazole-mediated cholestasis by use of in vitro methods. In addition, the applicability of NAMs for hazard identification in a regulatory context to detect specific forms of drug- or substance-induced liver injury (DILI/SILI) was considered. Therefore, we selected the two triazoles Pi and Te as test substances in in vitro assays by using the human liver cell lines HepG2 and HepaRG. On top of that we investigated the compounds in a binary mixture to see if the mixture response was in line with the assumption of concentration addition for substances with a similar mode of action. Besides BSEP inhibition, reporter gene assays were used to analyze nuclear receptor transactivation relevant for cholestasis. Additionally, we measured gene and protein expression as well as the level of mitochondrial membrane disruption. Results were finally compared to existing in vivo results for various azole fungicides used as pesticides or drugs, based on test guidelines generally applied in regulatory hazard assessment to allow the applicability of the NAM for regulatory purposes to be assessed. Concentration inducing effects in vitro were also compared to concentrations measured in vivo.

## 2. Materials and Methods

### 2.1. Chemicals

Technical grade Pi (CAS # 60207-90-1; Batch # CGA64250B; purity 96.10%) was purchased from Syngenta (Basel, Switzerland) and technical grade Te (CAS # 107534-96-3; Batch # NK21BX0392; purity 96.20%) from Bayer (Leverkusen, Germany). Dimethylsulfoxide (DMSO) was used as solvent for the test compounds and added to the cells resulting in a final DMSO concentration of 0.2% (*v*/*v*). Equimolar mixtures of Pi and Te were applied in mixture trials (i.e., “5 μM Pi+Te” corresponds to 2.5 μM Pi + 2.5 μM Te). Table 1 shows structural formulas of the used compounds.

### 2.2. Cultivation of HepaRG and HepG2 Cells

Undifferentiated HepaRG cells (Biopredic International, Saint Grégoire, France) were seeded and cultured as previously described [16,17]. In brief, cells were cultured in William’s E medium with 2 mM glutamine (Pan-Biotech, Aidenbach, Germany) supplemented with 10% fetal calf serum (FCS; Pan-Biotech), 100 U/mL penicillin and 100 μg/mL streptomycin (Capricorn Scientific, Ebsdorfergrund, Germany), 0.05% human insulin (PAA Laboratories GmbH, Pasching, Austria) and 50 μM hydrocortisone hemisuccinate (Sigma-Aldrich). After cultivation for two weeks, cells were differentiated in the medium mentioned above containing 1.7% DMSO in addition for another two weeks. Differentiated HepaRG cells were treated with test compounds in treatment medium (phenol red-free Williams E medium, Pan-Biotech, supplemented with the same supplements as the differentiation medium, but only 2% FCS and 0.2% DMSO) for 24 h or six days with renewal of medium/treatments every two days. Human HepG2 hepatocellular carcinoma cells (ECACC, Salisbury, UK) were cultured in Dulbecco’s modified Eagle’s medium (DMEM; Pan-Biotech) supplemented with 10% FCS (Pan-Biotech) as described previously [16]. Treatment with test substances was conducted in phenol red-free DMEM medium (Pan-Biotech) supplemented with 10% FCS for 24 h. A Binder cell culture incubator was used for incubation of both cell lines at 37 °C and 5% CO_2_ in a humidified atmosphere.

### 2.3. Cell Viability Tests

Cytotoxicity of Pi and Te was analyzed using the colorimetric MTT (3-(4,5-dimethylthiazol-2-yl)-2,5-diphenyltetrazolium bromide) reduction assay in HepG2 and HepaRG cells in 96-well format according to standard protocols [18]. The detergent Triton X-100 (0.01%) served as positive control. Concentrations of 40 μM (24 h treatment) and 20 μM (6 days treatment), yielding ≥80% cell viability, were chosen as the maximum concentrations of Pi and Te for further experimentation to ensure the absence of artifacts caused by cytotoxicity.

### 2.4. Gene Expression Analysis

Gene expression analysis was conducted as previously described [7,19] The human microarray Agilent Expression Profiling Service (incl. 8 × 60K Array) was conducted by ATLAS Biolabs GmbH (Berlin, Germany) as previously described [12,19] using RNA from HepaRG cells treated with a combination of 10 μM Pi and 10 μMTe for 24 h. Results (fold changes of each treatment relative to the solvent control, 0.2% DMSO) were further evaluated using the bioinformatics analysis and search tool IPA (Ingenuity Pathway Analysis) from QIAGEN (Germantown, MD, USA) using the IPA “Tox Analysis” tool. A *p*-value < 0.05 and a |fold change| > 2 were used as cutoff criteria for the transcriptomics data. In IPA, standard settings (no filtering, direct as well as indirect relationships were considered) were selected (date of analysis: 12 October 2018) [12]. PCR-based gene expression analysis was conducted as recently described [7,19] using the primer pairs listed in Appendix A.

### 2.5. Plasmids and Dual Luciferase Reporter Assay

Dual luciferase reporter gene assays for CAR, PXR and FXR were conducted to analyze the capability of Pi and Te to activate these nuclear receptors in HepG2 cells. The plasmids and assays have been described in detail before [16]. In brief, HepG2 cells were cultivated in 96-well plates and transiently transfected with plasmids (Appendix A) using TransIT-LT1 (Mirus Bio LLC, Madison, WI, USA) in a relation of 3:1 (TransIT-LT1 [μL]: amount of plasmids [μg]). For the FXR transactivation assay, the first plasmid is based on a fusion protein of GAL4 with the ligand-binding domain (LBD) of FXR. The second plasmid contains a firefly luciferase reporter gene under control of the GAL4-specific upstream activation sequence (UAS). For the CYP7A1 promoter assay, the luciferase reporter construct is driven by a fragment of the promoter of the human FXR-responsive *CYP7A1* gene [14]. All measurements were performed according to the Dual Luciferase Assay protocol as provided by the manufacturer (Promega, Madison, WI, USA) and detailed elsewhere [19,20] by using a plate reader (Infinite M200PRO, Tecan, Männedorf, Switzerland).

### 2.6. Bile Salt Export Pump Assay

The assay was conducted by Solvo (Szeged, Hungary) according to standard protocols. In brief, it utilizes isolated cell membrane preparations from HEK293 cells overexpressing human BSEP. The probe substrate was incubated in the presence of ATP (active transport condition) or AMP (passive diffusion condition) in triplicates (n = 3). Control incubations with a known BSEP substrate (taurocholate), as well as taurocholate together with the known BSEP inhibitor cyclosporine A, were conducted alongside the incubations.

### 2.7. Mass-Spectrometric Determination of Transport Protein Expression

Expression levels of transport proteins were investigated using a so-called Triple X Proteomics (TXP) targeted proteomic analysis as described in a previous study [15]. In essence, cell pellets were lysed in buffer for one hour, and the protein concentration was subsequently determined by the bicinchoninic acid (BCA) assay (Thermo Fisher Scientific, Waltham, MA, USA) according to the manufacturer’s manual. Next, proteolysis was performed overnight with trypsin. Endogenous as well as stable isotope-labeled reference peptides were precipitated using TXP antibodies (customized production by Pineda, Berlin, Germany) using magnetic beads coated with protein G (Thermo Fisher Scientific). The precipitated peptides were then quantified using a previously described 10 min LC-MS (liquid chromatography coupled to mass spectrometry) method with an UltiMate 3000 RSLCnano and a tSIM-QExactive Plus™ mass spectrometer (Thermo Fisher Scientific). Peak areas of the known amounts of the isotope-labeled peptides were set in relation to endogenous signals at the parent ion level [21].

### 2.8. Alteration of Mitochondrial Membrane Function

Another key event (KE) in the cholestasis AOP is the alteration of the mitochondrial membrane function. The JC-1-assay was used to address this KE via measurement of the voltage-dependent accumulation of the charged dye JC-1 (5,5′,6,6′tetrachloro- 1,1′,3,3′-tetraethylbenzimidazol-carbocyanine iodide). JC-1 enters the mitochondria and accumulates depending on the membrane potential. For high membrane potential, JC-1 aggregates and emits red light (595 nm). Conversely, it emits green light (535 nm) at low membrane potential. For the measurements, HepaRG cells were cultivated in 96-well plates and treated with the test substances Pi and Te for six days. Valinomycin (10 μM) was applied for 24 h and served as positive control. A volume of 100μL JC-1-solution was added to the cells. After an incubation time of 20 min and washing twice with PBS, fluorescence emission was measured. The ratio of red/green (595 nm/535 nm) fluorescence values were calculated and finally normalized to the solvent control.

### 2.9. Comparison to In Vivo Results

In vivo results from previous studies of our group [7,8] as well as from regulatory guideline studies as summarized by Nielsen et al. in 2012 [10] were considered to evaluate whether cholestasis could be observed at the histopathological level in rodents following repeated-dose treatment with Pi and Te. Human in vivo drug side effects by azoles were obtained by searching the European drug vigilance database for the substances fluconazole, itraconazole and ketoconazole, at http://www.adrreports.eu/en/index.html (accessed on 8 June 2021).

### 2.10. Statistical Analysis

Statistical analysis was conducted with SigmaPlot for Windows software (Version 14.0). Shapiro–Wilks and Brown–Forsythe tests were respectively used to analyze if results were normally distributed and for homogeneity of variances. Since most of the data did not meet the prerequisites for parametric testing, the non-parametric Mann–Whitney rank sum test was applied. An asterisk (*) indicates statistical significance at *p* < 0.05, and error bars depict the standard deviation. IC50 values were calculated by linear regression from measured values with SigmaPlot for Windows software.

## 3. Results

In the course of an evaluation of microarray gene expression data from a previous study on the combined effects of Pi and Te in HepaRG human hepatocarcinoma cells, bioinformatic analyses indicated that the compounds exerted transcriptional changes potentially related to hepatic adverse outcomes (Figure 1b; see also Knebel et al., 2019 [12]). This approach was chosen as HepaRG cells constitute an in vitro system closely resembling human hepatocytes (see also discussion section for more information), and since our own previous analyses (e.g., see [12]) have proven the usefulness of bioinformatic analyses of omics data as a NAM to obtain information about liver toxicity and its mode of action. Of note, the majority of toxicity-relevant functions affected by the compounds, as determined by bioinformatic analysis, showed a close relationship to the activation of nuclear receptors. For example, predicted effects of Pi and Te such as liver carcinoma formation, hepatic steatosis, hyperplasia, and liver enlargement are prototypical consequences of CAR and PXR activation in liver cells. Interestingly, transcriptomic data also predicted hepatic cholestasis as an outcome of Pi and Te treatment of HepaRG cells. Liver cholestasis is an adverse outcome also related to nuclear receptor activation, including not only the activation of PXR and CAR (Figure 1a), but also the farnesoid-X-receptor (FXR). The present study therefore aimed to elucidate the potential of both compounds to induce cholestatic effects in human liver cells, as well as to perform comparative analyses in order to assess the in vivo relevance of the findings. The results are presented in accordance with the AOP proposed in Figure 1a. Therefore, NAMs were used to first check the MIE (BSEP inhibition), followed by reporter gene assays for the nuclear receptors involved. Subsequently, the expression of genes and proteins as well as the mitochondrial membrane function for the respective KE were checked. Finally, the results obtained by NAMs in vitro were compared to in vivo results from previous studies, including compound concentrations measured in vivo as reported in a previous publication [8].

### 3.1. BSEP Inhibition

Activity of BSEP was analyzed using an in vitro approach based on membrane preparations of human HEK293 cells as a NAM tailored specifically to detect interaction with this particular transport protein. We found that BSEP was inhibited by both compounds: Pi inhibited BSEP-mediated taurocholate accumulation in a concentration-dependent manner with a maximum inhibition of 91% at a concentration of 300 μM (Figure 2a). The calculated IC50 was 78.56 μM (Table 2). Te inhibited BSEP-mediated taurocholate accumulation in a concentration-dependent manner with a maximum inhibition of 99% at a concentration of 300 μM (Figure 2a). The calculated IC50 was 38 μM (Table 2). Both compounds should therefore be regarded weak to moderate inhibitors of BSEP, which generally is in line with the assumption that inhibition of BSEP is the MIE of the AOP for hepatic cholestasis. In addition, expression of the ABCB11 gene (encoding BSEP) was significantly down-regulated at the mRNA level in a concentration-dependent manner in HepaRG cells by Pi alone (significant at the highest dose level only) and by the equimolar mixture of Pi and Te (Figure 2b).

### 3.2. Nuclear Receptor Activation

According to the AOP, BSEP inhibition will lead to subsequent activation of the nuclear receptors PXR, CAR, and FXR (Figure 1a). This was analyzed using reporter gene assays in HepG2 cells, due to limitations of HepaRG cells with regard to transfectability. The reporter systems used as NAMs here measure the potential of fusion proteins of the respective receptor and the GAL4 protein, brought into the cells via transient transfection of plasmid DNA (e.g., see [12]). Activating binding of the test compound to the nuclear receptor will induce transcription of a luciferase reporter gene. Both compounds activated PXR (Table 1; Appendix A; see also methodology and data in Refs. [12,19]). On the other hand, Pi and Te exerted opposite effects on CAR, with Pi acting as an activator and Te as an inhibitor (Appendix A; see also methodology and data in Refs. [12,19]). Please note that data on PXR activation have been published previously [12]. With respect to FXR, statistically significant receptor activation was observed after Pi treatment, but not with Te (Figure 3a). Thus, in summary, PXR was consistently activated by both test compounds, whereas the effects on CAR (Pi inducer, Te inhibitor) and FXR (Pi inducer, Te no effect) were inconsistent.

### 3.3. Transcriptional Changes Related to Cholestasis

In line with assumptions from the AOP, Pi and Te provoked several alterations in the expression of genes associated with cholestasis in HepaRG cells. Alterations observed at the mRNA level are summarized in Figure 4b. For details please refer to Appendix A. Among these was a down-regulation of CYP7A1, the rate-limiting key enzyme in bile acid synthesis (for details see Figure 3c). This was confirmed using a luciferase reporter system driven by a fragment of the human *CYP7A1* promoter in HepG2 cells (Figure 3b). Additional transcriptional changes were observed e.g., for *CYP8B1*, which was down-regulated by Pi and Te, as well as for the up-regulated genes *ABCC3* and *ABCG5* (Figure 4a). CYP8B1 is an enzyme involved in bile acid synthesis, while the transporters ABCC3 and ABCG5 are involved in the cellular export of bile acids and cholesterol. While most of these effects were observed as an early response to exposure 24 h after the start of incubation, some were also observable after 6 days of treatment. Equimolar mixtures of both azoles led to transcriptional effects similar to the individual compounds. Figure 4b gives an overview of all genes affected that are responsible for bile acid excretion or synthesis. For some of the proteins encoded by the aforementioned genes, mass-spectrometric assays for protein determination were available. Thus, protein level confirmation of some of the observed changes was achieved (Figure 4b). Detailed protein data are presented in Appendix A.

### 3.4. Alteration of Mitochondrial Membrane Function

To assess possible disruption of the mitochondrial membrane, a further KE in the cholestasis AOP, the JC-1-assay was performed. This NAM is based on a fluorescent dye which accumulates in mitochondria depending on their membrane potential. We found that Pi and Te alone as well as their equimolar combination depolarize the mitochondrial membrane (Figure 5). Mitochondrial depolarization was presumably due to a disruption of the cellular organelle causing the mitochondrial permeability transition pores to open. In turn, the equilibrium of charges was gradually achieved and thus the membrane potential decreased. This decrease is in line with the assumption made in the AOP of cholestasis that mitochondrial function is altered.

### 3.5. In Vivo Cholestasis by Triazoles in Rodents

In order to allow for a comparison of in vitro and in vivo cholestatic effects of triazoles, published data were reviewed for cholestasis-relevant observations with triazoles applied in agriculture or as pharmaceuticals. For the pesticidal active substances Pi and Te, in vivo effects observed in rodent bioassays after short- and long-term oral exposure were hepatocellular hypertrophy and vacuolization, as well as slight increases in liver enzymes in clinical chemistry (for details see [9]). However, cholestasis was not observed in vivo in guideline-compliant repeated-dose toxicity studies for either of the two substances (see Table 1).

For other active substances from the azole group, a summary of in vivo studies was checked for toxicological findings by Nielsen et al., 2012 [10]: three active substances were identified that caused cholestasis in vivo, namely, difenoconazole, prothioconazole and triticonazole. However, the proportion of substances from the azole group causing cholestasis appears rather small, as there are at present more than 20 azole compounds used as active substances in agriculture.

### 3.6. In Vivo Cholestasis by Triazoles in Humans

For drugs, the European database on pharmacovigilance was checked for reports on the clinically used azole fungicides itraconazole, ketoconazole and fluconazole (www.adrreports.eu; accessed on 8 June 2021). For itraconazole, approximately 900 adverse drug reactions (ADRs) were reported affecting the hepatobiliary system in the ADR database until 8 June 2021. Of these, only 22 cases showed a cholestatic phenotype (cholestasis or cholestatic liver injury). Jaundice was observed in 56 cases, jaundice cholestatic in only 5 cases. Hyperbilirubinaemia was observed in 16 cases. For ketoconazole, approximately 350 adverse drug reactions were reported affecting the hepatobiliary system in the ADR database until 8th June 2021. Of these only 12 cases showed a cholestatic phenotype (cholestasis or cholestatic liver injury). Jaundice was observed in 37 cases. Hyperbilirubinaemia was observed in 3 cases. For fluconazole, approximately 1100 adverse drug reactions were reported affecting the hepatobiliary system in the ADR database until 8 June 2021. Of these approximately 110 cases showed a cholestatic phenotype (cholestasis or cholestatic liver injury). Jaundice was observed in 84 cases, jaundice cholestatic in 14 cases. Hyperbilirubinaemia was observed in 32 cases. Overall, these results indicate that some azoles generally have the potential to cause cholestasis in vivo in rodent bioassays or in humans when used as drugs for treatment of mycosis. However, in comparison to other hepatobiliary effects of azoles, cholestasis appears to occur at a relatively low frequency.

## 4. Discussion

In vitro transcriptomics analysis revealed cholestasis as one of the top pathways affected after treatment of human liver-derived HepaRG cells with Pi and Te. This is in line with transcriptomics findings for other azoles like cyproconazole, epoxiconazole or prochloraz in vitro and in vivo [7,13,22].

Considering the molecular AOP concept, the present observations with Pi and Te go well in line with the assumption of a cholestatic potential of azoles, as the MIE (inhibition of the activity of BSEP) was clearly affected, and also because effects on nuclear receptors were recorded, especially a consistent and robust activation of PXR by both test compounds. In addition, diminished expression of *ABCB11*—the gene encoding BSEP—was observed at the mRNA level, adding an additional aspect to the mode of action of azoles to interfere with the AOP for liver cholestasis. This is an interesting observation: the literature is not consistent regarding the effects of PXR, FXR and CAR ligands towards BSEP expression, reporting either induction or no substantial effects [23,24]. Therefore, the lowered levels of BSEP mRNA might point towards additional molecular mechanisms of Pi and Te beyond nuclear receptor activation.

However, no cholestasis was observed for Pi and Te in a number of in vivo studies summarized by Nielsen et al., 2012 [10]. This was also confirmed by the absence of such an adverse effect in a number of regulatory studies performed according to OECD test guidelines within the approval processes of the active substances in Europe (summarized in the respective EFSA conclusions [25,26]). Is this an indication that the NAM-based testing or the AOP concept may fail?

Several explanations related to the sensitivity of the used cell lines as compared to primary hepatocytes or the in vivo situation could be discussed in this context to explain the observed discrepancy, as could incomplete correlation between mRNA analyses and actual protein levels, as well as sensitivity of the applied methods. However, the HepaRG model has been compared to primary hepatocytes and in vivo models, and found to be of comparable sensitivity and functionality with respect to many liver-specific features [27,28]. Therefore, even if the above reasons cannot fully be ruled out as underlying causes, a different explanation seems much more likely.

Indeed, on the one hand, neither of the two azoles under investigation caused cholestasis in standard rodent bioassays in vivo. Considering this, the transcriptomics approach is suspect of being over-predictive. On the other hand, one cannot deny that substances of the azole fungicide group bear a certain potential to cause cholestasis, as shown for some other agricultural fungicides of the same class in vivo [10] and as also documented by cases of DILI, pointing towards a cholestatic potential of some azole fungicides used as human drugs (as documented in the ADR database and summarized in Section 3.5 above). The present results indicate that Pi and Te are both capable of inducing molecular changes in human liver cells, which are indicative of cholestasis, and which are in line with the assumptions of the respective AOP (e.g., BSEP inhibition, PXR activation). In order to judge the in vitro findings, it is crucial to consider different aspects of toxicodynamics and toxicokinetics, which might explain the apparent gap between the different findings mentioned above. These may be species differences, difficulties of in vitro–in vivo extrapolation or kinetic differences.

In general, species differences may be the underlying cause of a divergence between rodent in vivo data and results from in vitro experimentation with human cells [29,30]. For several azole fungicides, for example, a non-genotoxic mechanism of liver tumor induction via activation of the receptor CAR has been identified, the relevance of which for humans is controversial [29,30]. However, the fact that some azoles can act in a cholestatic manner in laboratory rodents and certain ones also in humans indicates that the chemical class of azoles generally has the capability of exerting cholestatic effects in both species, making a scenario of fundamental species differences for this particular endpoint appear rather unlikely.

A general in vitro–in vivo discrepancy for the ability of azoles to exert cholestatic effects in humans or rodents does also appear very unlikely, as the observation that azoles may cause DILI in human patients treated for mycosis is generally in line with the in vitro observations in cultured human liver cells. This has been observed in rare cases for itraconazole, ketoconazole or fluconazole as reported in the EU ADR database analyzed in Section 3.5. Moreover, a small proportion of the azole fungicides used in agriculture (e.g.,difenoconazole or prothioconazole) have shown a cholestatic potential in vivo [10]. In addition, the cell lines used have successfully been used by others on several cholestatic substances [3,31].

Instead, the main reason for the non-occurrence of cholestasis with Pi and Te in vivo seems to be kinetic in nature, and related to the cellular levels of the azole compounds achieved in the different types of experiments: as shown in Table 1, the IC50 for BSEP inhibition is ~5–50 fold above the intra-hepatic concentration measured at the top dose in previous animal studies [8]. Thus, assuming that the inhibition of murine BSEP by Pi and Te occurs at similar concentrations as for human BSEP, the tissue levels needed to induce BSEP inhibition in vivo have probably not been reached in the respective studies. Nuclear receptor activation, as exemplified by PXR activation and associated with other liver effects such as CYP induction, hypertrophy and hepatocellular proliferation, occurs in a dose range similar to the observed in vivo concentrations (see Table 1), explaining why the latter forms of nuclear receptor-mediated liver effects are observed in vivo. In line with the AOP for hepatic cholestasis, PXR activation alone appears not to be sufficient to trigger the adverse outcome, whereas BSEP inhibition is most likely needed.

Our results illustrate both strengths and weaknesses of in vitro-based NAM assays involving toxicogenomics. On the positive side, these methods accurately detect certain hazards and allow for screening of potential effects without the need for animal testing. On the other side, NAMs may be over-predictive, especially when aspects of kinetics are not considered.

It appears most likely that the potential of Pi and Te to induce cholestasis is, in general, correctly predicted by NAM in vitro indicating the usefulness of these NAMs. However, parameters such as in vivo tissue concentrations and compound potencies have to be considered carefully when using these data to predict adverse outcomes in vivo (as demonstrated in Table 1). Hence, if not used for definitive prediction of an adverse outcome on its own but either in combination with other approaches or for prioritization for further testing, the approach is well suited to fit into a tiered regulatory procedure as an early step to predict a hazard. In a later phase of such an approach, specific organ concentrations, known from in vivo studies or computed using appropriate toxicokinetic models including in silico models, need to be considered in order to conclude on the risk of an adverse outcome [32]. A number of in silico physiologically-based pharmacokinetic (PBPK) models exist that may in principle be used, as recently reviewed by Ref. [33].

In the present study, this was not the case as the concentrations needed for in vitro inhibition of BSEP exceed the tissue concentrations achievable in rodent in vivo studies. In this context it should be mentioned that the concentrations triggering cholestasis in vitro are also several orders of magnitude above realistic human exposure levels, as maximum residue levels in food are well below the effect dose levels.

## 5. Conclusions

The present work demonstrates the ability of NAM-based in vitro testing involving toxicogenomics to correctly identify hazardous properties of substances. It also shows the necessity to combine such methods with considerations of toxicokinetics to come to sound conclusions in risk assessment.

## Figures and Tables

**Figure 1 cells-11-03293-f001:**
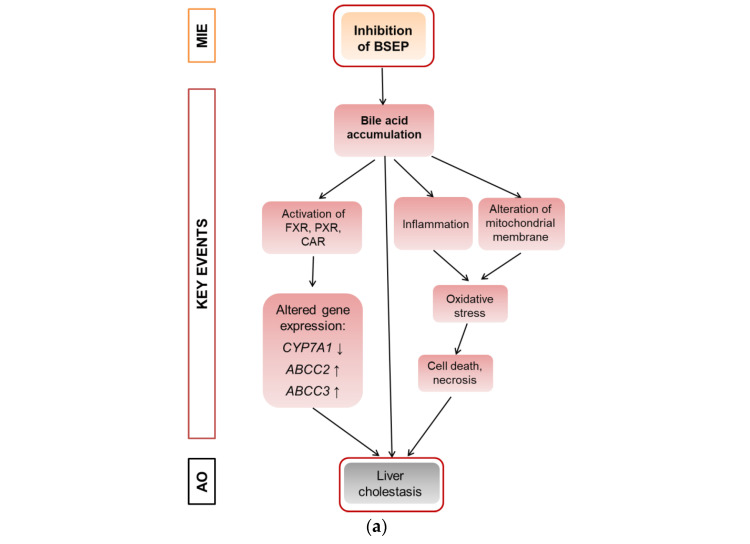
Adverse outcome pathway (AOP) for liver cholestasis and transcriptomic evidence for cholestatic effects of propiconazole (Pi) and tebuconazole (Te) in human liver cells. (**a**) Schematic representation of an AOP for liver cholestasis modified after a previous publication [2]. Inhibition of the bile salt export pump (BSEP) as the molecular initiating event (MIE; orange) triggers several key events (red) that in turn lead to the adverse outcome of liver cholestasis (grey). AO: Adverse outcome, BSEP: Bile salt export pump, FXR: farnesoid-X-receptor, CAR: constitutive androstane receptor, PXR: pregnane-X-receptor. (**b**) Bioinformatic analysis of transcriptome data indicates alterations in several hepatotoxicity-related pathways after treatment of HepaRG cells with a mixture of Pi and Te (10 μM each), including liver hyperplasia and hyperproliferation, liver steatosis, and in this specific case, liver cholestasis. The most prominently affected pathways (“Tox Functions”) are depicted, as identified using IPA (Ingenuity Pathway Analysis) software. Data analysis was performed using transcriptomics data from a previous study [12].

**Figure 2 cells-11-03293-f002:**
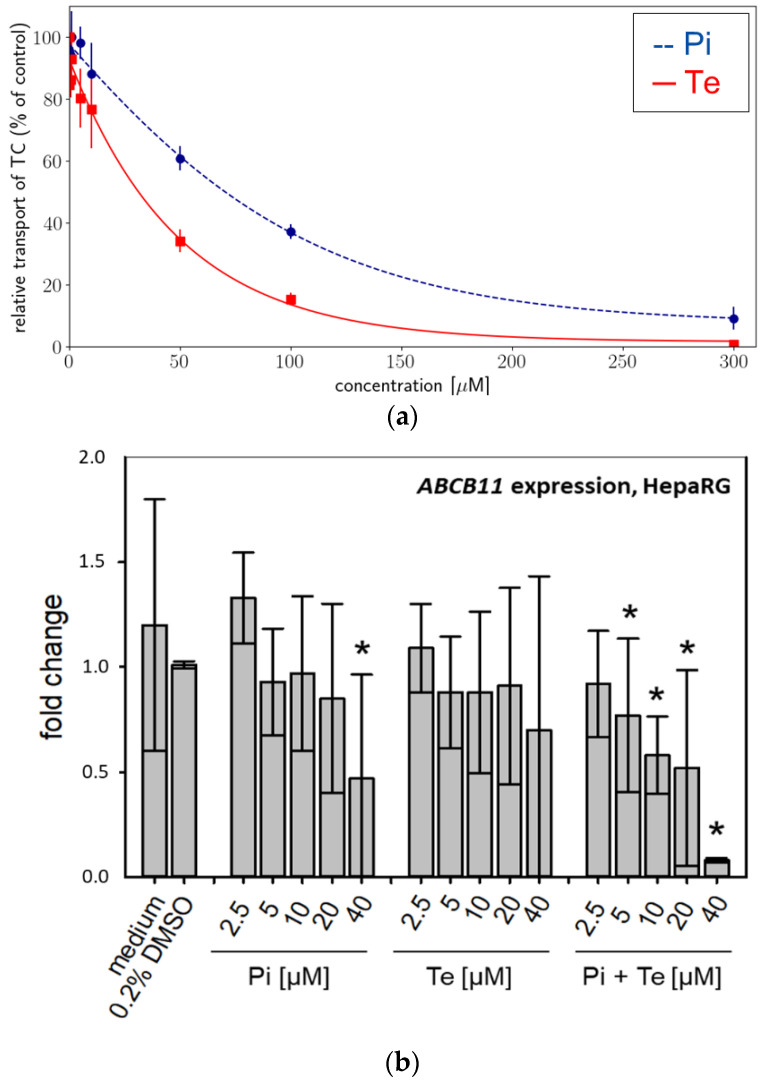
Effects on bile salt export pump (BSEP) activity and expression. (**a**) In vitro activity of BSEP in HEK293 cell membrane preparations in the presence of increasing concentrations [0–300 μM] of propiconazole (Pi; blue dashed line) or tebuconazole (Te; red solid line) expressed as relative taurocholate (TC) transport against the solvent control. (**b**) Gene expression of ABCB11 (the gene encoding BSEP) after 24 h of treatment with Pi, Te, and their equimolar combination (Pi + Te) from 2.5 μM to 40 μM in HepaRG cells. Statistical significance was reached at *p* < 0.05 and is indicated by asterisks (Mann–Whitney rank sum test against the solvent control).

**Figure 3 cells-11-03293-f003:**
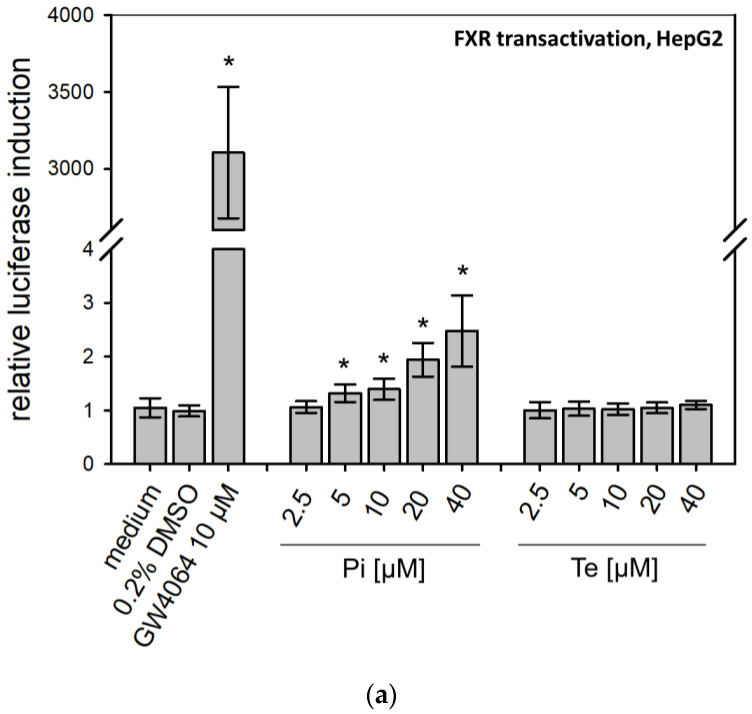
Effects of propiconazole (Pi) and tebuconazole (Te) on farnesoid-X-receptor (FXR) transcriptional activity and its target *CYP7A1*. (**a**) Dual luciferase reporter transactivation assay for FXR in HepG2 cells, incubated with the test substances for 24 h. Relative induction of a luciferase-based reporter gene system based on a fusion protein of GAL4 with the ligand-binding domain of human FXR is shown after treatment with Pi, Te, and their equimolar combination from 2.5 μM to 40 μM. (**b**) CYP7A1 promoter assay in HepG2 cells. Activities of a luciferase-based reporter construct driven by a section of the promoter of the human *CYP7A1* gene are shown as compared to solvent control after treatment with Pi, Te, and their equimolar combination from 2.5 μM to 40 μM for 24 h. (**c**) CYP7A1 expression in HepaRG cells after 24 h of treatment with Pi, Te, and their equimolar combination (Pi + Te) from 2.5 μM to 40 μM for 24 h. Statistical significance, reached at *p* < 0.05, was tested by Mann–Whitney rank sum test against the solvent control and is indicated by asterisks.

**Figure 4 cells-11-03293-f004:**
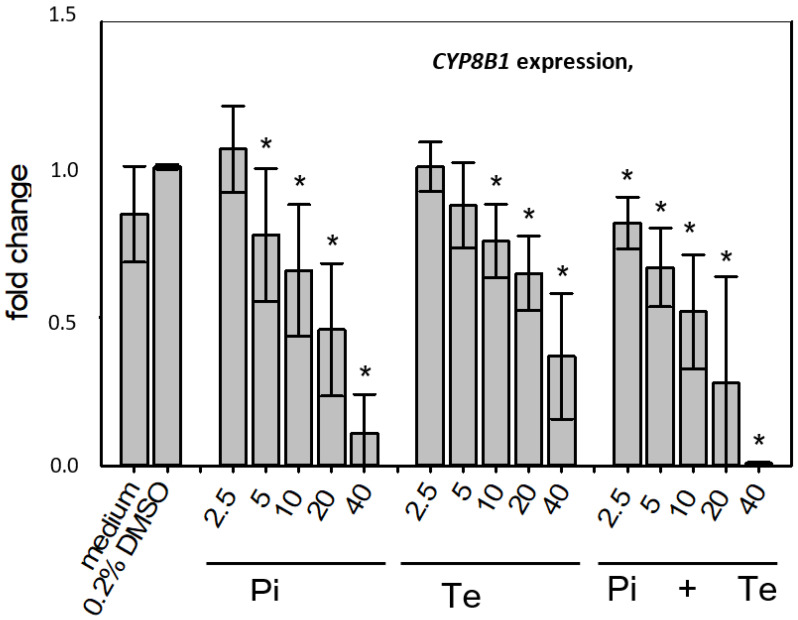
Cholestasis-relevant gene expression changes caused by propiconazole (Pi), tebuconazole (Te), or their combination. (**a**) CYP8B1, ABCC3 and ABCG5 expression in HepaRG cells after 24 h of treatment with Pi, Te, and their equimolar combination (Pi + Te) from 2.5 μM to 40 μM. Statistical significance was reached at *p* < 0.05 and is indicated by asterisks (Mann–Whitney rank sum test against the solvent control). (**b**) Overview of effects of Pi (blue) and Te (red) on the expression of cholestasis-related genes. HepaRG cells were treated with Pi or Te for 24 h. Upward and downward arrows indicate up- or down-regulation if (i) statistical significance was reached at *p* < 0.05 (testing as described in the methods section), and (ii) fold change of expression was ≥1.25 or ≤0.75. * indicates that the respective gene was also altered at the protein level. Abbreviations: ABC: ATP binding cassette; CYP: cytochrome P450; SLC: solute carrier.

**Figure 5 cells-11-03293-f005:**
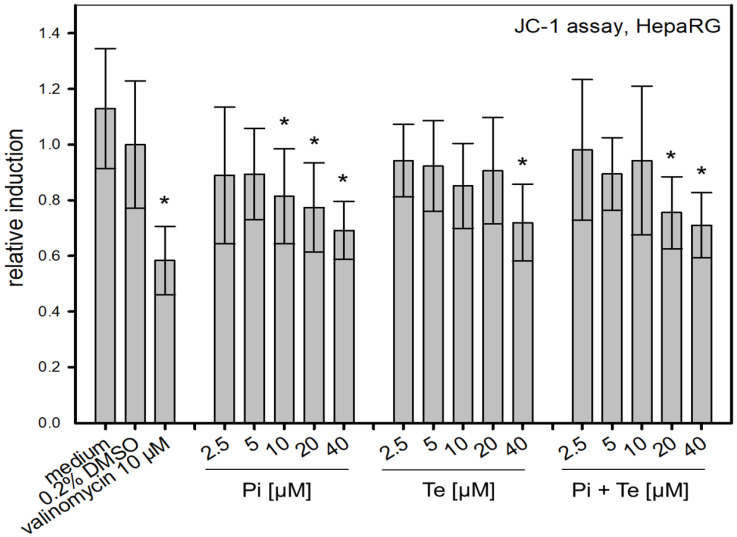
Alteration of the mitochondrial membrane function measured via the JC-1-assay. Mitochondrial disruption is caused by propiconazole (Pi), tebuconazole (Te), and their combination in HepaRG cells after 6 h of treatment from 2.5 μM to 40 μM. The relative induction was calculated based on the solvent control (normalized for the value measured for 0.2% DMSO). Statistical significance was reached at *p* < 0.05 and is indicated by asterisks (Mann–Whitney rank sum test against the solvent control). Valinomycin (10 μM) was applied for 24 h and served as positive control.

**Table 1 cells-11-03293-t001:** Structural formulas of propiconazole (Pi) and tebuconazole (Te).

Propiconazole (Pi) CAS # 60207-90-1	Tebuconazole (Te) CAS # 107534-96-3
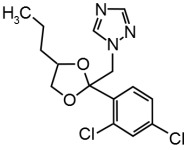	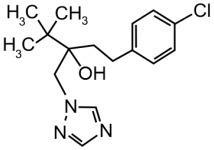

**Table 2 cells-11-03293-t002:** Concentrations of Pi and Te found to activate the pregnane-X-receptor (PXR) responsible for enzyme induction, to inhibit the bile salt export pump (BSEP) responsible for induction of cholestasis, measured in vivo concentrations, and observed in vivo effects.

Substance	EC_50_ PXR Transactivation [12]	IC_50_ BSEP	In Vivo Liver Conc. At Top Dose [8]	Liver Effects [8]
Pi	~6 μM	79 μM	1.3 μM	Enzyme induction but no cholestasis
Te	~8 μM	38 μM	5.9 μM	Enzyme induction but no cholestasis

## Data Availability

The datasets used and/or analyzed during the current study are available from the corresponding author upon reasonable request.

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
