# Peer review of "New Approach Methods for Hazard Identification: A Case Study with Azole Fungicides Affecting Molecular Targets Associated with the Adverse Outcome Pathway for Cholestasis"

_cells, 2022, doi:10.3390/cells11203293_

Round 1

Reviewer 1 Report

The study by Knebel et al. investigates the effects of the two fungicides propiconazole (Pi) and tebuconazole (Te) to induce cholestasis using NAMs/ human in vitro liver cell systems (HepaRG and HepG2).

The study is well designed and performed and analysed critically. Overall the manuscript is written very well. Results reported here are important to be published and will help to further develop NAMs. I only have some minor comments that should be addressed.

-          In the methods (2.4) please report more details. What cut off were used for IPA analysis? Values of fold changes and p-adjust values should be reported. Which statistical methods was employed to test for multiple testing (False discovery rate)?

-          Figure 2b, please revise the text in the main manuscript. The expression levels are only changed in a concentration-dependant manner for the combined treatment. For single treatment it was not significant for Te and only at the highest concentration significant for Pi.

-          Section 3.4 uses often present tense. Change to past tense

-          Section 3.6 often a space is missing (lines 385,394, other)

-          Figure 5, can the authors explain what exactly is meant by “relative induction” on the y-axis in the Figure legend? Is this fluorescent ratio of red/green (595nm/535nm) fluorescence values of treated dived by ratio of red/green (595nm/535nm) fluorescence values of solvent control?  Was this measured by plate reader or fluorescence microscopy? Could you add a representative image of the red and green fluorescence?

-          The discussion about the concentration differences between concentrations tested in vitro and in vivo ( 5 to 50 fold) is of great importance, regarding the potential over-prediction of NAMs. Can the author comment on the expected maximum concentration that may be expected in humans? Furthermore, cancer cell lines are employed here, do the authors expect them to be more sensitive compared to for example primary human hepatocytes?

-          Another point to add to the manuscript is that here mainly mRNA expression levels were reported in HepaRG cells and that these may not always reflect functional activity of transporter or CYP enzymes ( even though function was evaluated in reporters or HEK model).

Author Response

Reviewer 1

The study by Knebel et al. investigates the effects of the two fungicides propiconazole (Pi) and tebuconazole (Te) to induce cholestasis using NAMs/ human in vitro liver cell systems (HepaRG and HepG2).

The study is well designed and performed and analysed critically. Overall the manuscript is written very well. Results reported here are important to be published and will help to further develop NAMs. I only have some minor comments that should be addressed.

Answer: We are grateful to Reviewer 1 for the comprehensive evaluation of our manuscript. We have addressed all comments by this reviewer (see detailed answers below).

-          In the methods (2.4) please report more details. What cut off were used for IPA analysis? Values of fold changes and p-adjust values should be reported. Which statistical methods was employed to test for multiple testing (False discovery rate)?

      Answer: We now report the cutoff criteria for p-value and fold change in the methods section, as recommended.

-          Figure 2b, please revise the text in the main manuscript. The expression levels are only changed in a concentration-dependant manner for the combined treatment. For single treatment it was not significant for Te and only at the highest concentration significant for Pi.

Answer: Thanks for this hint. The text has been changed accordingly.

-          Section 3.4 uses often present tense. Change to past tense

Answer: The use of past tense has been improved as suggested.

-          Section 3.6 often a space is missing (lines 385,394, other)

Answer: We have added missing spaces.

-          Figure 5, can the authors explain what exactly is meant by “relative induction” on the y-axis in the Figure legend? Is this fluorescent ratio of red/green (595nm/535nm) fluorescence values of treated dived by ratio of red/green (595nm/535nm) fluorescence values of solvent control?  Was this measured by plate reader or fluorescence microscopy? Could you add a representative image of the red and green fluorescence?

Answer: An explanation for relative induction has been added to the figure legend.

-          The discussion about the concentration differences between concentrations tested in vitro and in vivo ( 5 to 50 fold) is of great importance, regarding the potential over-prediction of NAMs. Can the author comment on the expected maximum concentration that may be expected in humans? Furthermore, cancer cell lines are employed here, do the authors expect them to be more sensitive compared to for example primary human hepatocytes?

Answer: The authors agree that the expected concentration in humans under realistic conditions of exposure is important as well as the sensitivity of the cell lines. The discussion has been improved as suggested.

-          Another point to add to the manuscript is that here mainly mRNA expression levels were reported in HepaRG cells and that these may not always reflect functional activity of transporter or CYP enzymes (even though function was evaluated in reporters or HEK model).

Answer: The authors agree to this remark and have amended the discussion accordingly.

Reviewer 2 Report

This manuscript investigated the potential of triazole fungicides like (propiconazole and tebuconazole) to induce cholestasis in human liver cells in vitro and further explored the underlying mechanism of cholestasis by genomics. In addition, the authors discussed approaches to genetics and in vitro models as the potential ability to study contaminants to cause specific types of toxicity. However, there are some errors presented in the manuscript, the manuscript is not suitable for publication in its present form, major revision is needed. The reviewer suggest that the authors describe the new approach methods in the background section. The logic of the article is flawed. First, the line 71 mentioned that the new approach methods have the advantage of avoiding animal testing. However, the line 486 also pointed out that the new approach methods need to be combined with toxicokinetics to obtain reasonable results. It is generally known that animal testing is required for toxicokinetics. Then what is the advantage of the new approach methods? 

Author Response

Reviewer 2

This manuscript investigated the potential of triazole fungicides like (propiconazole and tebuconazole) to induce cholestasis in human liver cells in vitro and further explored the underlying mechanism of cholestasis by genomics. In addition, the authors discussed approaches to genetics and in vitro models as the potential ability to study contaminants to cause specific types of toxicity. However, there are some errors presented in the manuscript, the manuscript is not suitable for publication in its present form, major revision is needed. The reviewer suggest that the authors describe the new approach methods in the background section. The logic of the article is flawed. First, the line 71 mentioned that the new approach methods have the advantage of avoiding animal testing. However, the line 486 also pointed out that the new approach methods need to be combined with toxicokinetics to obtain reasonable results. It is generally known that animal testing is required for toxicokinetics. Then what is the advantage of the new approach methods?

Answer: We thank Reviewer 2 for the evaluation of our manuscript. We have performed a major revision of the paper and clarified that with respect to toxicokinetics, also in silico models could be used (PBTK modeling). This is especially important as some governments responsible for setting the data requirements for approval of chemical substances, such as the US or the Dutch governments, have indicated to phase out animal testing by 2030.

Additionally, the text has been amended to better describe the new approach methods, as suggested by this reviewer.

Reviewer 3 Report

This is a very interesting article combining research and review,the results illustrate both, strengths and weaknesses of in vitro-based NAM assays involving toxicogenomics. NAMs may be overpredictive, especially when aspects of kinetics are not considered.Tthe necessity to combine such methods with considerations of toxicokinetics to come to sound conclusions in risk assessment should be given enough attention.It is very practical and instructive.

Main question 1: the author proposed the importance of analysis combined with pharmacokinetic results. Can some suitable pharmacokinetic screening methods and means be mentioned in the discussion, which will be more instructive.

2.Pharmacokinetic differences involve many factors, but the structural differences of compounds are the basic factors. It is suggested that the author add a table to list the structural formulas of each azole compound exemplified in the article, which is more conducive to the readers' future expansion and analysis.

Details:

1.Line 130,[7,16,there should be a bracket missing.

2.Line 213,andTe,there should be a bracket space missing.

3.Page 6,Figure1a,in the notes, the meanings of MIE and AO abbreviations should be added. In addition, it is suggested that the links between the three stages on the left side of the figure and the contents on the right side should be closer. Maybe it would be better to a little change or add appropriate connecting lines.

4. Page 9,Figure 3(b),there is no label for the test item.

Author Response

Reviewer 3

This is a very interesting article combining research and review,the results illustrate both, strengths and weaknesses of in vitro-based NAM assays involving toxicogenomics. NAMs may be overpredictive, especially when aspects of kinetics are not considered.Tthe necessity to combine such methods with considerations of toxicokinetics to come to sound conclusions in risk assessment should be given enough attention.It is very practical and instructive.

We are grateful to Reviewer 3 for the comprehensive evaluation of our manuscript. We have addressed all comments by this reviewer (see detailed answers below).

Main question 1: the author proposed the importance of analysis combined with pharmacokinetic results. Can some suitable pharmacokinetic screening methods and means be mentioned in the discussion, which will be more instructive.

Answer: Pharmacokinetic methods are now mentioned in the discussion section as suggested.

  1. Pharmacokinetic differences involve many factors, but the structural differences of compounds are the basic factors. It is suggested that the author add a table to list the structural formulas of each azole compound exemplified in the article, which is more conducive to the readers' future expansion and analysis.

Answer: Thank you. Structural formulas have been added as suggested.

Details:

1.Line 130,[7,16,there should be a bracket missing.

Answer: Thank you. Amended as suggested.

2.Line 213,andTe,there should be a bracket space missing.

Answer: Thank you. Amended as suggested.

3.Page 6,Figure1a,in the notes, the meanings of MIE and AO abbreviations should be added. In addition, it is suggested that the links between the three stages on the left side of the figure and the contents on the right side should be closer. Maybe it would be better to a little change or add appropriate connecting lines.

Answer: Thank you. Abbreviations are now explained.

  1. Page 9,Figure 3(b),there is no label for the test item.

Answer: Thank you. Amended as suggested.

Round 2

Reviewer 2 Report

All my concerns have been well addressed by the authors, thank you for the efforts. However, in Figure 4a, the vertical coordinates (fold change) are occluded by the image. Therefore, the manuscript is not suitable for publication in its present form, revision is needed.

Author Response

We would like to thank Reviewer for his carefull revision and for noting the occlusion of the coordinates in Fig 4a. We have now re-formated the figure and the coordinates should now be viewable.

We also did a comprehesive spell check and hope that the manuscript is now suitable for publication.